# Ageing and brain white matter structure in 3,513 UK Biobank participants

Simon R. Cox[1,2,3,*], Stuart J. Ritchie[1,2,*], Elliot M. Tucker-Drob[4], David C. Liewald[1,2], Saskia P. Hagenaars[1,2,5], Gail Davies[1,2], Joanna M. Wardlaw[1,3,6], Catharine R. Gale[1,2,7], Mark E. Bastin[1,3,6] & Ian J. Deary[1,2]

Quantifying the microstructural properties of the human brain's connections is necessary for understanding normal ageing and disease. Here we examine brain white matter magnetic resonance imaging (MRI) data in 3,513 generally healthy people aged 44.64–77.12 years from the UK Biobank. Using conventional water diffusion measures and newer, rarely studied indices from neurite orientation dispersion and density imaging, we document large age associations with white matter microstructure. Mean diffusivity is the most age-sensitive measure, with negative age associations strongest in the thalamic radiation and association fibres. White matter microstructure across brain tracts becomes increasingly correlated in older age. This may reflect an age-related aggregation of systemic detrimental effects. We report several other novel results, including age associations with hemisphere and sex, and comparative volumetric MRI analyses. Results from this unusually large, single-scanner sample provide one of the most extensive characterizations of age associations with major white matter tracts in the human brain.

[1] Centre for Cognitive Ageing and Cognitive Epidemiology, University of Edinburgh, Edinburgh EH8 9JZ, UK. [2] Department of Psychology, University of Edinburgh, Edinburgh EH8 9JZ, UK. [3] Scottish Imaging Network, a Platform for Scientific Excellence (SINAPSE) Collaboration, Edinburgh EH8 9JZ, UK. [4] Department of Psychology, University of Texas, Austin, Texas 78712-0187, USA. [5] Division of Psychiatry, University of Edinburgh, Edinburgh EH10 5HF, UK. [6] Brain Research Imaging Centre, Neuroimaging Sciences, Centre for Clinical Brain Sciences, University of Edinburgh, Edinburgh EH4 2XU, UK. [7] MRC Lifecourse Epidemiology Unit, University of Southampton, Southampton SO17 1BJ, UK. * These authors contributed equally to this work. Correspondence and requests for materials should be addressed to S.R.C. (email: simon.cox@ed.ac.uk).

Fully understanding brain ageing requires an accurate characterization of how and where white matter microstructure varies with age. White matter is highly relevant to ageing: later-life cognitive decline may partly be caused by cortical disconnection, a microstructural deterioration of the brain's connective pathways through processes such as axonal demyelination that reduces information transfer efficiency[1–3]. The concept of disconnection has in large part been supported by analyses of diffusion magnetic resonance imaging (dMRI), a non-invasive, quantitative method that exploits the Brownian motion of water molecules, allowing inferences to be made about the underlying microstructure of brain white matter *in vivo*[4–6].

Past findings have been inconsistent in their characterization of the trajectory and spatial distribution of age effects across brain white matter tracts and of associations with hemisphere and biological sex[2,7–23]. A statistically well-powered study of age associations with white matter microstructure, in particular in middle-aged and older age groups (45 + years), would address these gaps in our understanding. In addition to conventional measures of fractional anisotropy (FA; the directional coherence of water molecule diffusion) and mean diffusivity (MD; the magnitude of water molecule diffusion), newer and rarely studied neurite orientation dispersion and density imaging (NODDI)[24] measures offer new information on the microstructural bases of age effects on white matter[25]. NODDI provides estimates of neurite density (intra-cellular volume fraction; ICVF), extracellular water diffusion (isotropic volume fraction; ISOVF) and tract complexity/fanning (OD). The observed mean decline of FA in older age could be affected by, among other factors[5], decreases in the density and/or an increase in the dispersion orientation of neurites (dendrites and axons); FA is unable to differentiate between these possibilities[26]. Thus, NODDI may offer a novel mechanistic insight into white matter ageing. However, a comprehensive examination of NODDI parameters has not been undertaken in the context of older age and has not been attempted alongside more commonly used water diffusion parameters.

The diffusion properties of white matter tracts across the brain are correlated; for example, an individual with relatively high FA in one tract is likely to also have relatively high FA across other tracts in the brain. This means that a latent, general factor of white matter microstructure can be derived[16,27]. Analysis of latent factors allows a deeper understanding of the covariance structure of inter-individual differences in age-related brain changes; measuring global white matter diffusion is of great interest for investigating ageing trends that are general across white matter tracts (but specific to white matter tissue)[10,11,14,21,28]. Isolated, tract-specific enquiry cannot distinguish whether ageing patterns are unique to that tract or an outcome of a more systemic constellation of processes. Tract-specific enquiry is also susceptible to a large degree of noise in the diffusion signal[29]; this noise can be reduced by using multivariate, latent-variable analyses[13].

An additional gap in our understanding relates to the de-differentiation hypothesis: the suggestion that interindividual differences in microstructure across tracts become increasingly related in older age[30–32]. This hypothesis stems from a common-cause theory of ageing-related neurodegeneration[33]. It posits that system-wide breakdown of physiological function shifts overall levels of integrity across brain regions for affected individuals. Given that these sources of variance are shared across brain tracts, the signature should be higher correlations among regionally distributed neuronal measures with age. Here we seek evidence for white matter structural de-differentiation from middle to older age, which could be an important marker of an aggregation of deleterious effects operating on white matter connections distributed across the central nervous system.

We undertook an analysis of age associations in 27 major white matter tracts (Fig. 1). The data were from the UK Biobank resource (http://www.ukbiobank.ac.uk). We examined the tract-specific differences according to age, sex and hemisphere. We conducted analyses across all five water diffusion measures discussed above (FA, MD, ICVF, ISOVF and OD) along with supplementary analyses of axial ($\lambda$ax) and radial diffusivity

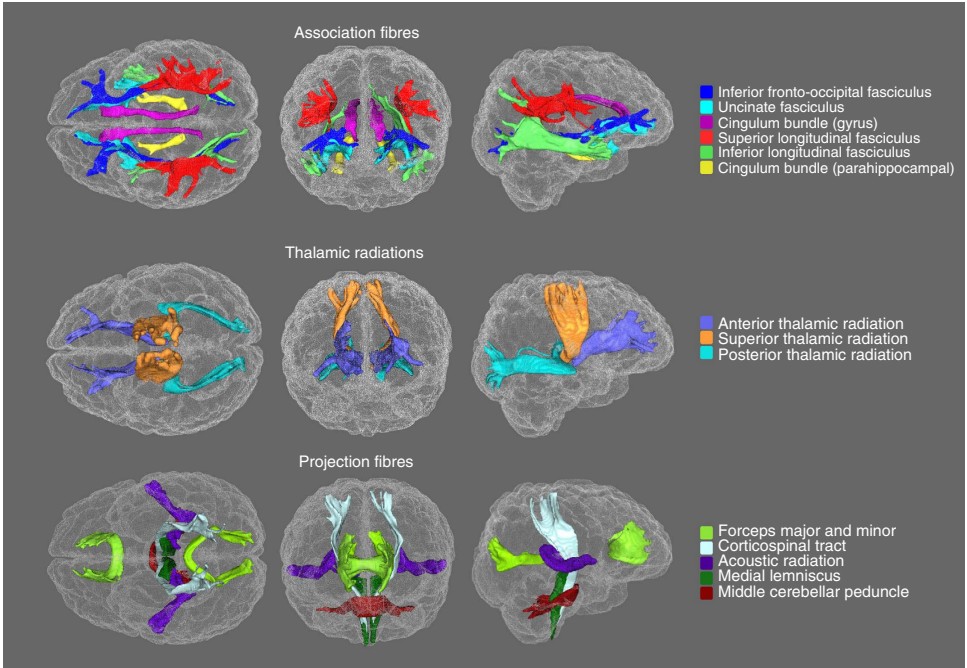

**Figure 1 | White matter tracts of interest.** Generated using probabilistic tractography rendered in superior (left), anterior (centre) and lateral (right) views.

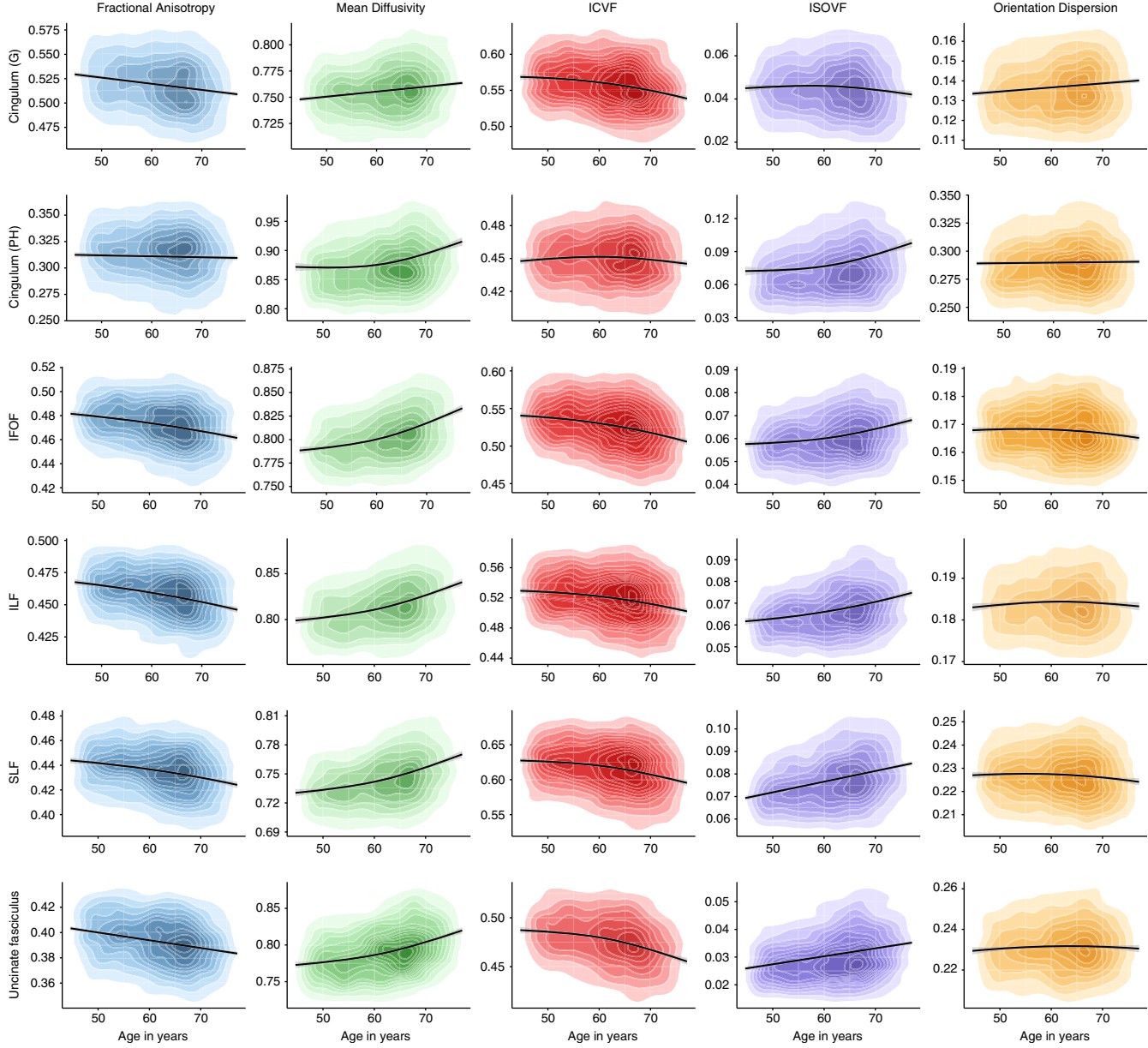

**Figure 2 | Age associations with the microstructural characteristics of association fibres.** Kernel density plots indicate the degree of data point overlap (darker = greater); black line denotes the linear or quadratic regression line (with grey 95% CIs) across the five microstructural measures. Blue, FA; green, MD; red, intracellular volume fraction; purple, ISOVF; orange, OD. G, gyrus; IFOF: inferior fronto-occipital fasciculus; ILF, inferior longitudinal fasciculus; PH, parahippocampal; SLF, superior longitudinal fasciculus.

(λrad), and diffusion tensor mode (MO), characterizing which biomarker and which tract was most age-sensitive. We investigated whether tracts' diffusion characteristics were more strongly correlated in older individuals (testing brain microstructural de-differentiation).

This study provides the most definitive characterisation to-date of age associations with the brain s white matter from middle to older age. We identified MD as the diffusion parameter most sensitive to age and, although sex differences exist, neither sex appeared to exhibit stronger age effects. Tract-specific associations with age were strongest in thalamic and association pathways, and weakest in projection fibres. We also show that there are tract-specific and white-matter-wide correlations with age. Finally, we found novel evidence for brain microstructural de-differentiation for FA, MD, ICVF and ISOVF.

## Results

**Tract characteristics.** Characteristics of the 3,513 participants and a recruitment flowchart are reported in Supplementary Table 1 and Supplementary Fig. 1. The current sample—the first release of the UK Biobank MRI sample—is a group of generally healthy middle-aged and older adults (age range 44.64–77.12 years). Tract-averaged values for MR diffusion parameters FA, MD, ICVF, ISOVF and OD in each brain pathway are displayed in Supplementary Fig. 2 and Supplementary Tables 1 and 2. We also report supplementary analyses of λax and λrad (both are of interest in ageing research but are similar to MD), and MO, which are each described in Methods. Tract-averaged values for these parameters are shown in Supplementary Fig. 3 and Supplementary Table 3. Associations among the left and right diffusion parameters are displayed in Supplementary Figs 4 and 5.

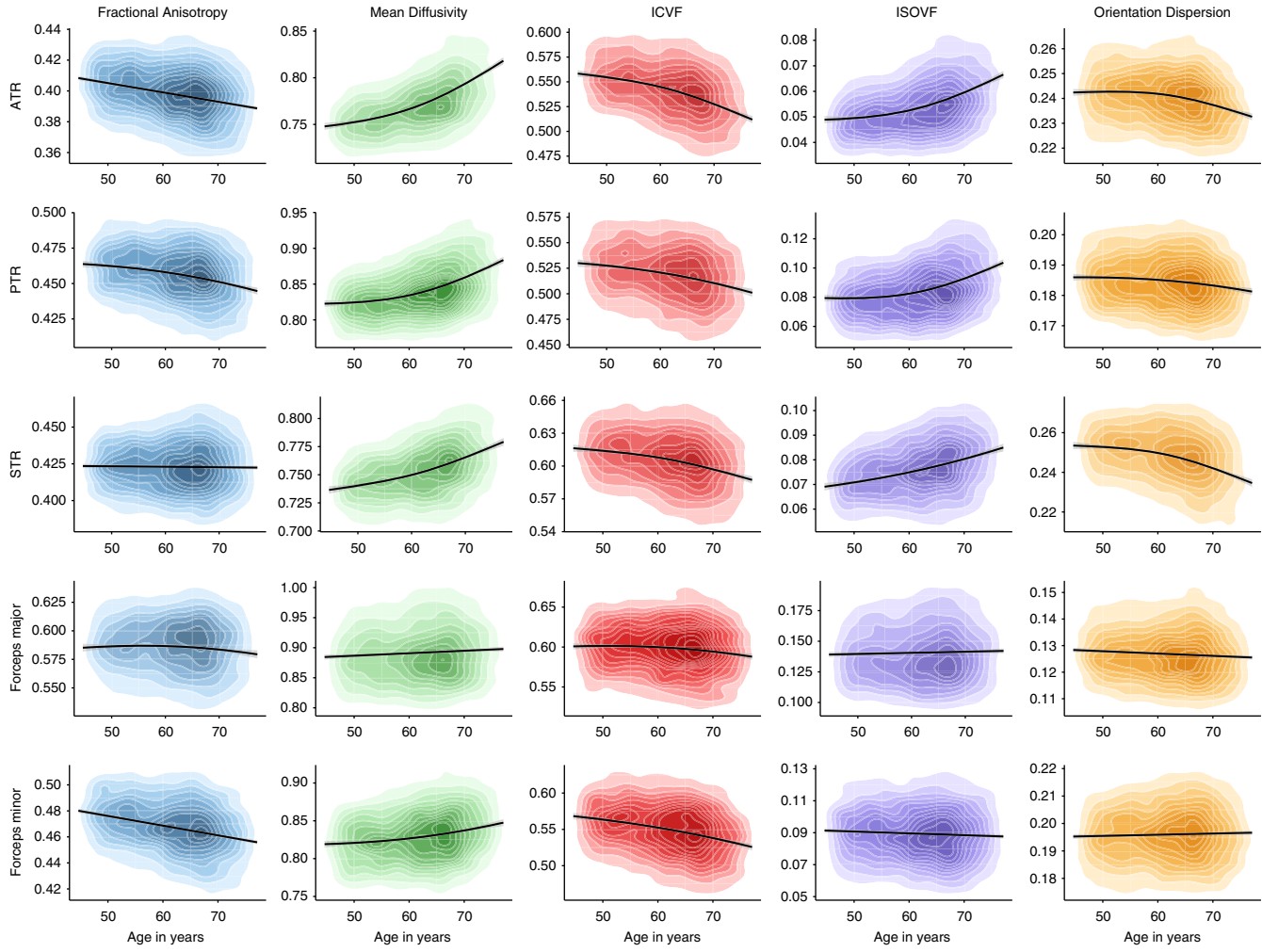

**Figure 3 | Age associations with the microstructural characteristics of thalamic and callosal fibres.** Kernel density plots indicate the degree of data point overlap (darker = greater); black line denotes the linear or quadratic regression line (with grey 95% CIs) across the five microstructural measures. Blue, FA; green, MD; red, intracellular volume fraction; purple, ISOVF; orange, OD. ATR, anterior thalamic radiation; PTR, posterior thalamic radiation; STR, superior thalamic radiation.

**Tract associations with age and hemisphere and sex**. Older age was significantly associated with lower coherence of water diffusion (FA; $\beta \geq -0.275$), lower neurite density (ICVF; $\beta \geq -0.382$) and lower tract complexity (OD; $\beta \geq -0.277$), and with a higher magnitude of water diffusion (MD; $\beta \leq 0.496$) and ISOVF ($\beta \leq 0.343$) across the majority of tracts (Figs 2–4 and Supplementary Tables 4 and 5). These results indicate less healthy white matter microstructure with older age. Both $\lambda$ax ($\beta \leq 0.478$) and $\lambda$rad ($\beta \leq 0.468$) showed comparable age associations to MD across all tracts and MO was associated both positively and negatively with age across tracts ($\beta$ range $-0.129$ to $0.212$; Supplementary Table 6). Associations were predominantly non-linear with the exception of FA and MO (Figs 2–4 and Supplementary Tables 4–6), indicating steepening slopes with increasing age. Associations with age were particularly marked in association (inferior fronto-occipital fasciculus (IFOF), inferior longitudinal fasciculus (ILF), superior longitudinal fasciculus (SLF) and Uncinate) and thalamic radiation (anterior thalamic radiation (ATR), superior thalamic radiation (STR) and posterior thalamic radiation (PTR)) fibres, as well as in the forceps minor (FMin) (Fig. 5). In contrast, the cingulum and sensory projection fibres showed modest or absent age associations across all

diffusion parameters. Differences in the magnitude of associations with age between these thalamic and association fibres (IFOF, ILF, SLF, Uncinate, ATR, STR and PTR) were consistently and significantly greater than those for the cingulum bundles, corticospinal tract (CST), middle cerebellar peduncle (MCP), medial lemniscus (ML), forceps major (FMaj), acoustic radiation (AR; Williams's one-sample $t$-values $> 2.42$, $P$-values $< 0.015$, $n > 3510$) across FA, MD, ICVF and ISOVF, with the exception of the cingulum bundle (gyrus) for ICVF. Among these specific association and thalamic radiation fibres, MD exhibited the strongest age associations of any of the diffusion parameters (Williams's one-sample $t$-values $> 4.27$, $P$-values $< 0.001$, $n > 3,510$; MD differences with $\lambda$ax and $\lambda$rad not tested; Fig. 6).

In addition to differential tract associations with age, putatively healthier microstructure was found for some measures in the left versus right hemisphere, such as higher FA ($\beta \geq -0.511$), lower ISOVF ($\beta \leq 0.319$) and lower MD ($\beta \leq 0.175$; Supplementary Tables 4–6 and Fig. 5). Males also showed consistently higher FA ($\beta \leq 0.218$). Females exhibited consistently greater OD ($\beta \geq -0.255$). The interactions between age and sex—although significant for some tracts (age × sex $\beta_{absolute}$ range = 0.028–0.079)—were small and inconsistent.

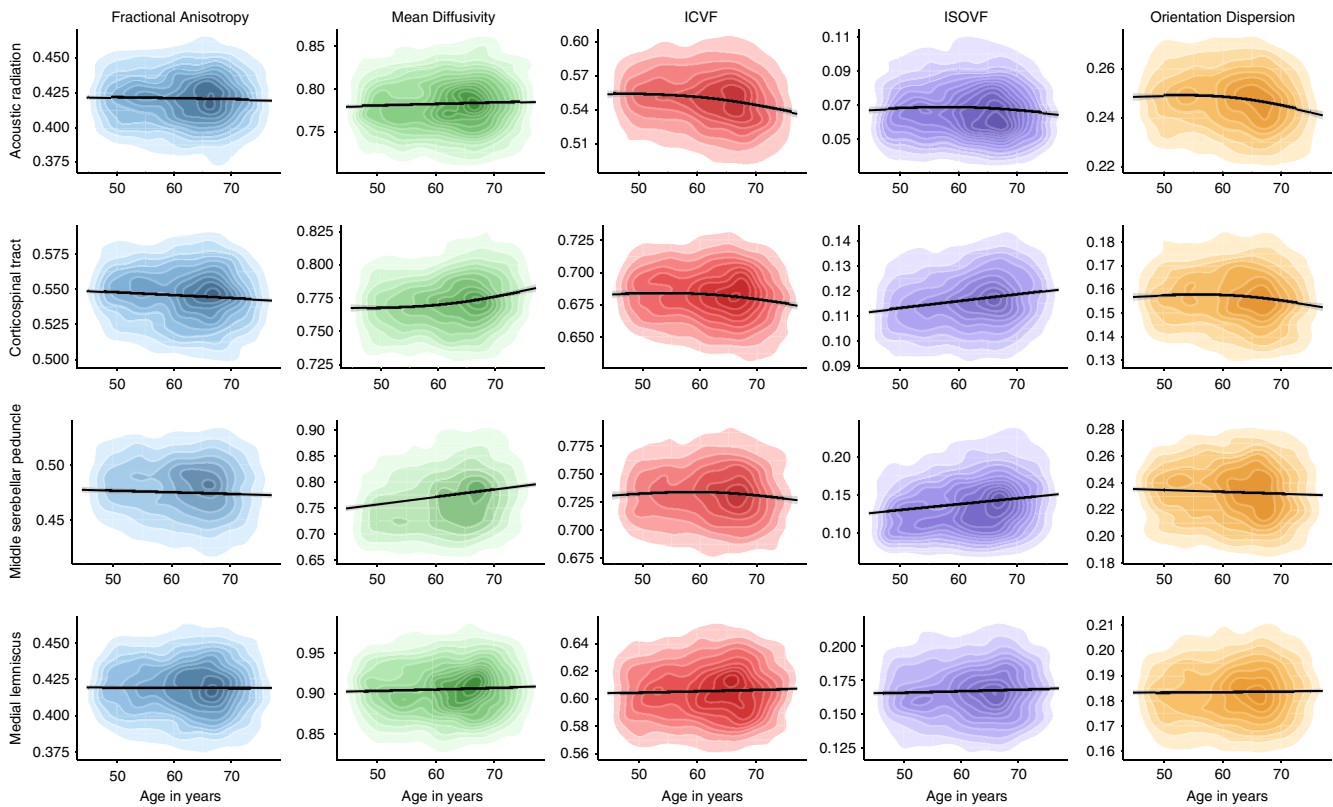

**Figure 4 | Age associations with the microstructural characteristics of sensory projection fibres.** Kernel density plots indicate the degree of data point overlap (darker = greater); black line denotes the linear or quadratic regression line (with grey 95% CIs) across the five microstructural measures. Blue, FA; green, MD; red, intracellular volume fraction; purple, ISOVF; orange: OD.

**General factors of white matter microstructure.** We tested whether there was evidence for latent factors explaining a substantial portion of the variance in each of the five different types of tract measurement. That is, we tested whether a given microstructural measure was positively correlated among all tracts across the brain and whether this common variance could be indexed by a latent factor.

Within all white matter biomarkers (with the exception of MO), the measurements from across all tracts correlated positively; for instance, those with higher FA in one tract tended to have higher FA in all their tracts (Fig. 7 and Supplementary Figs 4 and 5). The MCP and the bilateral cingulum gyrus, parahippocampal and ML tracts consistently covaried relatively weakly with others; these were removed from further models. The latent factors were thus indicated by 22 tracts each. For FA, MD, ICVF and ISOVF, initial scree plots of the tract data (Fig. 8, left panel) provided evidence for a strong single factor capturing common variance across the tracts; this was less clear for OD, which had a comparatively weaker first factor and a stronger second factor than the other measures. Examination of scree plots for λax, λrad and MO indicated similarly strong evidence for a strong first factor for λax and λrad, but extremely weak evidence for MO (Supplementary Fig. 6, left panel). As a consequence, a factor score for MO was not extracted and analysed further. Across the age range, the first factor accounted for a mean of 41.4% of total variance in FA, 38.1% in MD, 29.8% in λax, 39.6% in λrad, 68.2% in ICVF, 30.8% in ISOVF and 20.1% in OD. Results presented below are therefore based on single factor models of each of these diffusion measures. Henceforth, the prefix g denotes these latent factors (for example, the general FA factor is denoted gFA). Fit statistics, factor loadings and residual covariance paths are shown in Supplementary Tables 7–9.

Age associations in the latent factors are illustrated in Fig. 8 (central panel and Supplementary Fig. 6 for gλax and gλrad). As expected from the individual-tract data discussed above, gFA, gICVF and gOD factors were lower in older age. gFA and gICVF showed relatively linear declines, and gOD showed decline after approximately age 60 years. gMD, gλax, gλrad and gISOVF showed substantial increases from 45 years of age. The standardized effect sizes for age were gFA: $\beta = -0.254$, gMD: $\beta = 0.368$, gICVF: $\beta = -0.265$, gISOVF: $\beta = 0.273$, gOD: $\beta = -0.120$, gλax: $\beta = 0.341$ and gλrad: $\beta = 0.363$ (Supplementary Table 10). Age had a significantly stronger association with gMD than with any of the other latent factors (all Williams's t-values > 5.55, all P-values < 0.0001, all n > 3,510; λax and λrad not tested).

We next examined the extent to which the effect of age on white matter is common to all tracts, or tract-specific. We tested whether age associations with the individual tracts were accounted for by the association between age and the general factor (a common pathway), or whether there were incremental tract-specific age associations (common plus independent pathways). For all latent white matter microstructural measurements, there was evidence of age being associated with the general factor. There were also additional tract-specific effects, that is, common plus independent pathway models fit significantly better than models that only included a common pathway of age associations (Supplementary Table 11). In summary, age appears to affect the white matter both overall and in some additional, specific tracts (Supplementary Fig. 7).

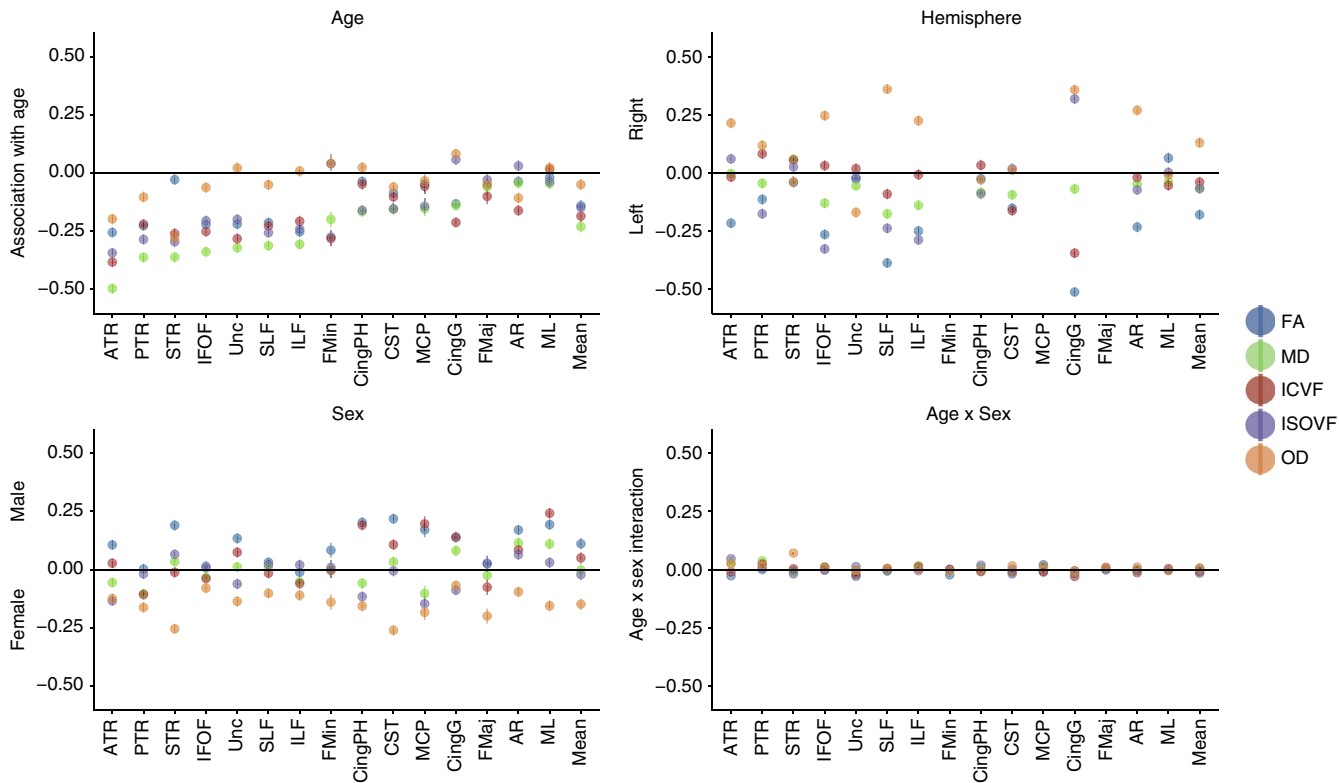

**Figure 5 | Associations between diffusion parameters with age, sex and hemisphere.** FA (blue), MD (green), ICVF (red), ISOVF (purple) and OD (orange) with age, sex and hemisphere. Error bars = 95% CIs. Female and left hemisphere coded as 0. The valence of MD and ISOVF associations have been reflected for the purposes of visualization for all four panels. See Supplementary Tables 3 and 4 for regression coefficients.

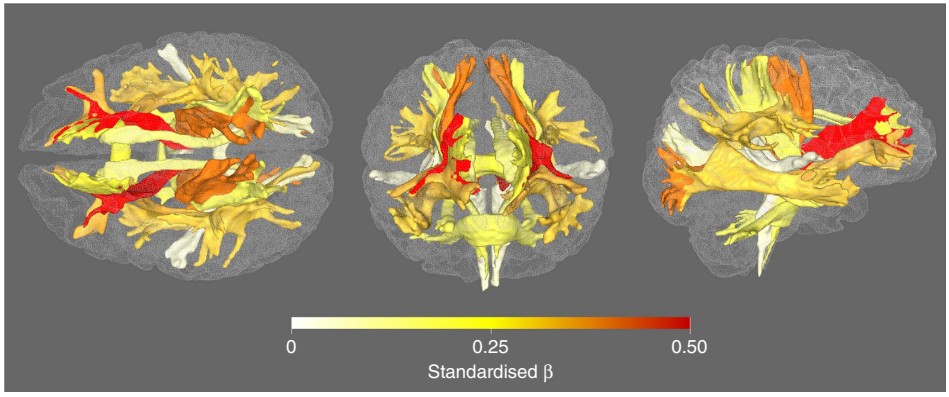

**Figure 6 | Association magnitudes of age and tract averaged MD.** Rendered in superior (left), anterior (centre) and lateral (right) views. Coefficient values (standardized βs) are from linear components of models shown in Supplementary Tables 3 and 4.

The FA signal is susceptible to microstructural properties in the brain that include neurite density and tract complexity/fanning. The availability of the NODDI variables ICVF and OD allow us to directly test whether and how ICVF and OD mediate the association between increasing age and lower FA. We used estimates of the general factor scores in a multiple-mediator analysis. Our results (Supplementary Table 12 and Supplementary Fig. 8) showed that the linear association between age and gFA was significantly mediated by 75.71% (from $\beta = -0.247$ to $\beta = -0.060$) with the inclusion of the NODDI parameters. The majority of this mediation took place through gICVF rather than gOD. This suggests that age-related declines in brain white matter FA may predominantly be explained by declines in neurite density rather than changes in tract complexity.

Next, we aimed to contextualize the utility of all five (FA, MD, ICVF, ISOVF and OD) white matter tract diffusion parameters' general factors in a reverse inference exercise, examining how much age variation they could explain beyond conventional brain volumetric measures in the same participants (total brain, grey matter, white matter, hippocampal and thalamic volumes; corrected for head size). Increasing age was associated with lower volumes in each of these measures (Supplementary Table 13 and Supplementary Fig. 9). Given the high collinearity of all volumetric and diffusion indices (Supplementary Table 14), we employed a penalized (elastic net) regression to identify an optimal set of predictors of chronological age in one half of the randomly and equally split sample (training set). $g\lambda$ax and $g\lambda$rad were virtually identical to gMD and were not included.

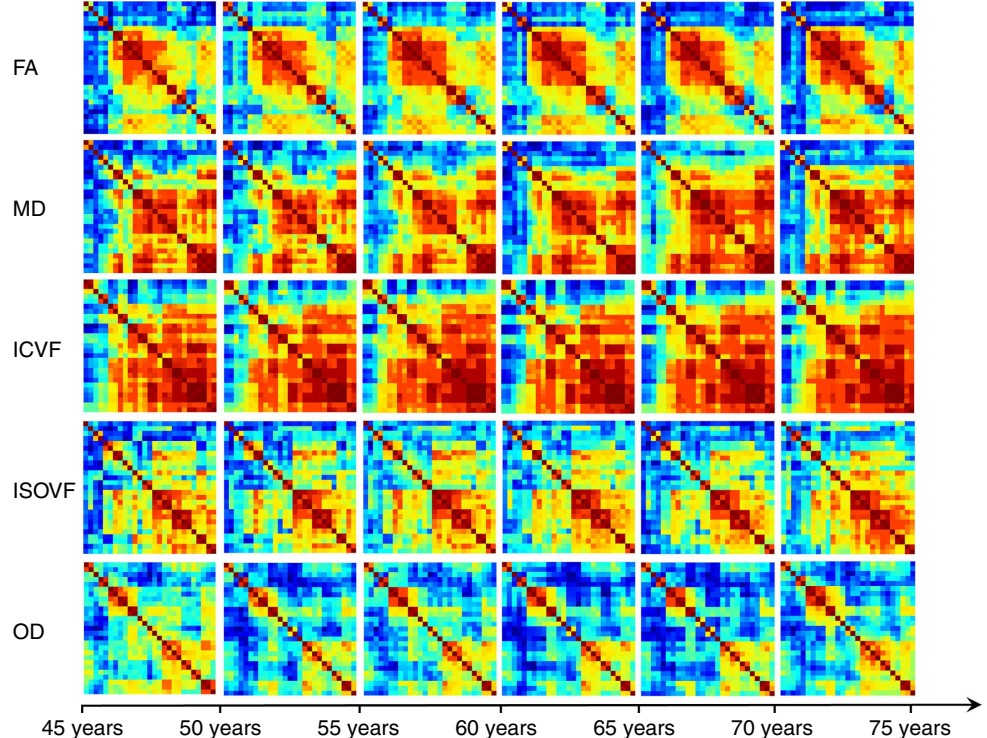

**Figure 7 | Illustrative heatmaps of tract de-differentiation for each parameter across age groups.** All tracts are shown. Higher tract inter-correlations are indicated by oranges and darker reds, with blues and greens denoting lower magnitudes.

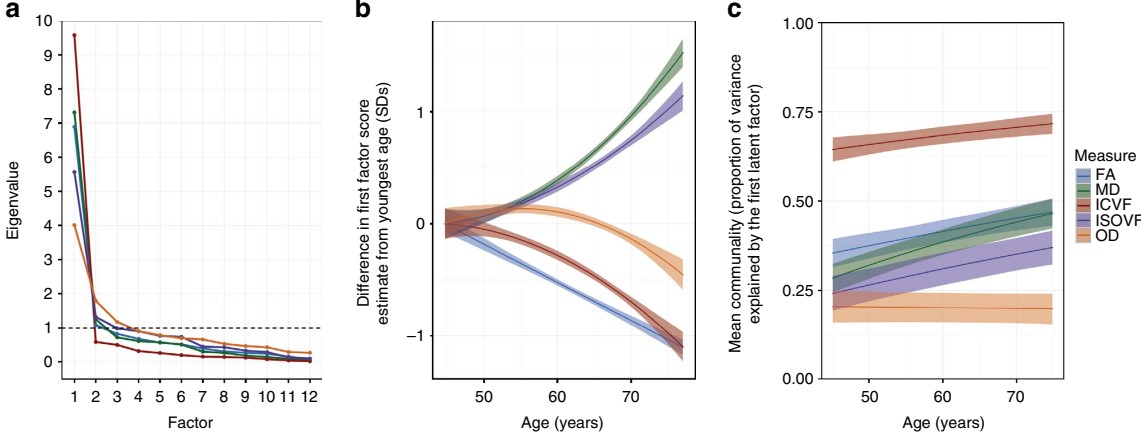

**Figure 8 | General factors of white matter microstructure explain greater variance with increasing age.** (**a**) Scree slopes for the exploratory factor analysis, showing the eigenvalue against the number of factors for each white matter tract measurement. (**b**) Age trajectories of the first (latent) factor of white matter microstructure for each of the five dMRI biomarkers. (**c**) Age de-differentiation of white matter microstructure. Age trajectories for the proportion of total variance in each tract measurement explained by the general factor. The shaded region around each trajectory shows ±1 s.d. of the mean.

The results showed that age-related variance in specific aspects of white matter microstructure (FA, MD and ISOVF) is partially independent of atrophy and grey matter volume. Of the five white matter microstructural measures and five volumetric measures, $g$FA, $g$MD, $g$ISOVF, total brain volume and grey matter volume appeared in more than 60% of the 1,000 bootstrapped models on the training set. These predictors were entered into a multiple linear regression in the training set ($n = 1,756$), followed by a confirmatory test using the same predictors in the testing set ($n = 1,757$; Supplementary Table 15). In both cases, $g$FA ($\beta_{\text{train}} = -0.085$ and $\beta_{\text{test}} = -0.071$, $P \leq 0.035$), $g$MD ($\beta_{\text{train}} = 0.085$ and $\beta_{\text{test}} = 0.109$, $P \leq 0.019$) and $g$ISOVF ($\beta_{\text{train}} = 0.119$ and $\beta_{\text{test}} = 0.124$, $P < 0.001$) accounted for

unique age variance, beyond total brain volume ($\beta_{\text{train}} = -0.160$ and $\beta_{\text{test}} = -0.171$, $P < 0.001$) and grey matter volume ($\beta_{\text{train}} = -0.375$ and $\beta_{\text{test}} = -0.348$, $P < 0.001$). For both models, these predictors accounted for nearly 40% of the age variance ($R^2 = 0.377$ in both cases) and did not exhibit multicollinearity (variance inflation factors $< 3.84$), indicating that the elastic net method had been effective in producing a useful set of predictor variables.

Given the apparent sensitivity of the thalamic radiations and of thalamic volume (Supplementary Table 13 and Supplementary Fig. 9) to age variance, we tested the degree to which the volume of the thalamus and a latent measure of thalamic radiation microstructure ($g$TR across each of the main five diffusion

measures) were uniquely informative of age, beyond general brain atrophy (total brain volume, corrected for head size). Results (shown in Supplementary Table 16) indicated that, in isolation, thalamic volume and $g$TR each significantly explained unique portions of age variance ($R^2$ for Thalamus + $g$TR FA = 0.129: Thalamus + $g$TR MD = 0.265; Thalamus + $g$TR ICVF = 0.175; Thalamus + $g$TR ISOVF = 0.218; Thalamus + $g$TR OD = 0.224). When total brain volume corrected for head size was included in the model, both thalamic volume ($\beta_{absolute} \leq 0.242$) and $g$TR for each diffusion parameter ($\beta_{absolute} \leq 0.229$) remained significant.

**Age de-differentiation of white matter microstructure.** Do white matter tracts tend to lose their individuality in older age? We tested the de-differentiation hypothesis that microstructural properties of white matter tracts across the brain become more similar at later ages. A series of heatmaps (Fig. 7) illustrate the increasing relatedness of white matter tract microstructure across six age groups with ~5 year intervals. Qualitatively stronger associations among tracts with increasing age were evident for FA, MD, ICVF and ISOVF, but not for OD. To quantify this more formally, we extended each of the general factor models to include an interaction parameter representing age moderation of the shared variance across the 22 tracts and an interaction parameter representing age moderation of the tract-specific unique variance. In other words, we estimated two interaction parameters for each of the diffusion measures. This allowed us, for each diffusion measure, to calculate the mean communality (the proportion of total variance that was shared) across tracts as a continuous function of age.

For six of the diffusion measures, the latent factor accounted for more variance in older age (Fig. 8 and Supplementary Fig. 6, right panel). For FA, the factor explained 11.5% more variance in the oldest participants (around age 75) than in the youngest participants (around age 45). The equivalent differences were 18.1% for MD, 11.8 for λax, 18.3% for λrad, 7.2% for ICVF and 12.9% for ISOVF. There was no appreciable age difference in the first factor of OD with age (there was a small, nonsignificant decline in explained variance of − 0.7%); we saw above that OD also evinced the weakest general factor overall. In summary, for six of the biomarkers, there was greater generality across the tracts with greater age, providing clear evidence of age-related de-differentiation. The older the brain, the more likely it will be that—within a given individual—one tract will share a microstructural quality with of all the brain's other white matter tracts. A dynamic illustration of white matter tract de-differentiation (using MD as an exemplar) can be seen in Supplementary Movie 1.

Finally, we evaluated whether the overall pattern of increasing tract communality with age was driven by particular subsets of the tracts. These analyses (Supplementary Tables 17–20 and Supplementary Figs 10–16) revealed that much of the upward mean trend in tract generality (for all measures except OD) shown in Fig. 8 was driven by steeper trends in the communalities of the association fibres and thalamic radiations—the same tracts that showed the largest mean age relationships. The communalities of the sensory projections (AR and CST) were flatter, indicating that the variance in them explained by the general factor was similar across the age range. Notably, some tracts (such as the FMaj) showed age differences in the amounts of variance explained at both the general and the specific level: that is, the total amount of tract variance was higher in older age.

Overall, we found that, at older ages, there was a greater tendency for all tracts in the brain to show similar levels of FA, MD, λax, λrad, ICVF and ISOVF within individuals. The increases in the tracts' common variance were particularly pronounced for the thalamic radiations and association fibres.

Therefore, it was not simply that age associations were largest among these specific tracts, but that interindividual differences in diffusion measures for these tracts covaried more strongly with each other and with other tracts in the brain in older individuals.

**Discussion**
This study adds to our understanding of the white matter microstructure of the brain in middle- and older-aged humans. Older age was most strongly associated with less healthy white matter in the thalamic radiations (ATR, PTR and STR) and association fasciculi (ILF, IFOF, SLF and Uncinate). The comparison of conventional dMRI and NODDI variables showed that MD was the most sensitive parameter to age among these tracts. Interindividual differences in each diffusion measure showed a clear tendency to covary across white matter tracts. General factors taken across the tracts captured substantial variation in FA, MD, ICVF and ISOVF, and more modest proportions of variation in OD. The general factors each had a tendency towards less healthy values at older ages and we also observed tract-specific age associations beyond those that could be accounted for exclusively via general factors.

Mediation models indicated that the link between older age and lower general FA ($g$FA) was predominantly driven by lower neurite density ($g$ICVF) rather than greater OD ($g$OD). In a reverse inference exercise, we illustrated the importance of brain diffusion parameters for understanding ageing: latent measures of white matter diffusion (FA, MD and ISOVF, in particular) provided unique information about age, beyond conventional volumetric brain measures of global atrophy, hippocampal, grey and white matter volume. We also showed that both the thalamic radiations and the volume of the thalamus itself are uniquely informative for age variance, beyond general atrophy. Finally, the tendency for white matter tracts within individuals to share microstructural qualities (assessed using water diffusion characteristics) was stronger in older participants, indicating de-differentiation of brain connectivity. This tendency was most strongly driven by the greater relatedness among the same thalamic radiation and association fasciculi that showed the greatest age associations in our tract-specific analyses.

The apparent differential age trends—both heterochronicity and spatial heterogeneity—across the tracts is supportive of the last in, first out hypothesis[34], whereby the tracts that are latest to develop are the most vulnerable to the deleterious effects of ageing. In particular, post-mortem and dMRI studies indicate ontogenetic differences between early-myelinating projection and posterior callosal fibres, and the later-developing association pathways[12,21,22,35]. An underlying reason for this pattern may be that the pathways latest to develop are more thinly myelinated, and that the oligodendrocytes responsible for their myelination are more vulnerable to aggregating deleterious effects due to their comparatively elevated metabolic activity[2,17]. We found that this pattern also occurs for more rarely studied water molecule diffusion measures (ICVF and ISOVF). Thus, these spatially distinct relations with age are apparent across a wider range of microstructural properties than was previously thought.

Our finding that the strongest age associations were with the thalamic radiations does not however follow the reported ontogeny in childhood. Whereas the anterior limb of the internal capsule (which contains, among other fibres, the ATR) shows a relatively delayed maturational trajectory, the posterior limb of the internal capsule (which contains, among other fibres, the PTR) shows comparatively early development along with the cortisospinal tract, ARs and MCP[36,37], yet the PTR showed some of the most marked age effects in this sample. However, the initial supposition of last in, first out referred to both ontogenetic and

phylogenetic chronology[34]; the thalamic nuclei are likely to have undergone considerable evolutionary modification, perhaps to keep pace with the rapidly expanding cortex[38]. The complex set of nuclei that comprise the thalamus share connections across the whole cortex, including hippocampal and prefrontal pathways, forming a densely interconnected processing unit[39–41], which may be highly relevant to its metabolic activity. The current study highlights the potential importance of the anterior, superior and PTRs (along with the complementary measure of the volume of the thalamus itself) and association fibres for understanding brain ageing; these tracts could be the foci of future investigations into behavioural outcomes and possible determinants.

Associations between each microstructural measure and age were generally in the expected direction, adding considerable confidence to the magnitude and (non-)linearity of prior estimates of decreasing FA, ICVF along with increasing MD, $\lambda$ax, $\lambda$rad and ISOVF from middle to older age across voxel-wise, region of interest and tract-based approaches[7,8,12,20–22,24,35,42]. Although our data do not cover childhood and earlier adulthood, the linear trajectories of FA reported here fit with prior evidence that FA declines with age may begin well before age 45 years, with MD relatively preserved until this age[9,12,21]. However, the negative association between age and gOD were not in line with some prior research[25,42], which found that increasing age was associated with increasing OD. The apparent discrepancy between studies might be ascribed to the following factors. We reported that gOD encapsulated less tract-wide variance than other general factors, reflected in the range of both positive and negative associations with age at the level of specific tracts. Thus, comparability with prior observations should be considered in light of the specific tracts considered. Moreover, our analysis showed that OD generally described a nonlinear increase until around 60 years, followed by a decrease (in some tracts more than others). Previous studies that did report an age-related increase in OD studied younger participants than in the current sample ($\leq 63$ years[25,42]), whereas a negative relationship, or a similar quadratic shift from positive to negative at ~60 years, was found when studying older participants[43,44]. All studies were also conducted using relatively small sample sizes (range $n = 47$–116). These results further emphasize the value of the current data, in which older participants are well-represented.

Age-related differences in white matter diffusion measures were not simply reflective of gross volumetric brain indices in our sample. All diffusion and volumetric measures were, in isolation, associated with age, but our analysis identified that grey matter volume, total brain volume and gISOVF, gFA and gMD were significant predictors (together explaining ~40% of the variance in age), whereas white matter volume, hippocampal volume, gICVF and gOD were not. The result that information about white matter water diffusion from both NODDI and more conventional measures is more informative for age than white matter volume might indicate that these biomarkers are more sensitive to subtle age-related differences in white matter. However, it is important to note that a global white matter measure can be divided into normal-appearing and white-matter hyperintensity volume in older subjects[45]. The latter, a marker of white matter disease[45], appears to progress independently of grey matter changes[46], but was not derived in the current data set. It may be that the insensitivity of total white matter volume to the important ratio of normal-appearing:white-matter hyperintensity volume may partly account for its lack of predictive value in our models. In addition, the current investigation focussed on specific pathways, therefore excluding peripheral white matter microstructure, which may also prove to be of particular interest to the brain and cognitive ageing[47]. The use of whole tract-averaged measures also assumes that the water diffusion

parameters are broadly equivalent throughout each tract of interest. Although this may be true for MD, FA reportedly exhibits local (within-tract) variability and larger differences between young and older participants (group $n \approx 50$) than MD[22]. Although another study reported that voxel-wise and tract-based analyses of white matter diffusion (including NODDI measures) showed concordant age associations[42], it is possible that methods such as tract-based spatial statistics (TBSS) may provide additional insight into locus-specific age effects within tracts (see refs 15,21), albeit in more limited portions of cerebral white matter.

The participants in the current study were in relative good health; we excluded individuals with any known neurodegenerative disease. Whereas there is evidence that MD is also more sensitive to age than FA among those with mild stroke[48], our sample composition has implications for another aspect of our results. Whereas hippocampal volume and ICVF did not appear to provide unique information about age in this sample, both ICVF (rather than FA)[49] and hippocampal volume[50] are important dementia biomarkers, emphasizing that our findings, which focus on normal ageing, cannot necessarily be generalized to clinical populations.

From each of the five white matter parameters, we extracted latent general factors to be extracted from the covariances among 22 white matter tracts. These factors indexed from 20 to 68% of the variance across the tracts, highlighting their importance—but not exclusivity—for providing indices of overall brain microstructure. The fit of these models was improved by adding correlated residual paths beforehand and we removed three fibre pathways for having low factor loadings. Thus, these are not the only factors that were extractable from the data: smaller, weaker factors may also exist that index appreciable portions of cross-tract variance[16]. Consequently, whereas we show that the association between age and gFA was predominantly attenuated by gICVF rather than gOD, we note that the latter did not exhibit such a strong first factor and therefore it is possible that our estimate of the amount that gOD contributes to the age-gFA relation might fluctuate more as a function of tract than it might for gICVF.

In our analysis of de-differentiation in the brain, we found compelling evidence—in four of the five white matter microstructural measurements (all except OD)—for increasing tract generality with age. For each of these four microstructural properties, the covariation across tracts within the brain was higher with greater age. One explanation of this de-differentiation is that the ageing of white matter structure involves an aggregation of systemic, detrimental effects that render tract-specific variation in white matter (dis)connectivity less prominent at older ages. Uncovering the specific cellular and metabolic mechanisms that might cause this increasing generality and how this generality might shed light on the brain basis for cognitive and functional ageing should be investigated in future studies. The large sample size provided crucial improvements in statistical power over previous studies. Based on our power calculations, even with a sample size of 1,000, we would have mischaracterized the vast majority of tract–age associations as linear, the modest age × sex interactions as null and, most importantly, we would most probably have lacked the precision to differentiate the various parameters' sensitivity to age and to accurately quantify the magnitude of white matter dedifferentiation.

Further to limitations of our sample composition and the brain MRI and statistical analyses discussed above, cross-sectional estimates of age-related trajectories may imperfectly index patterns of within-person changes over time[51]. They are also subject to bias due to cohort differences and secular trends in the brain structure. Nevertheless, longitudinal studies remain rare

and the time course required to test healthy participants prospectively across a comparable age span is prohibitive, leading many to adopt semi-longitudinal designs across large age ranges[8,10,16,23,52,53], with comparatively smaller samples across a wide age range, or adopt full longitudinal designs in cohorts with a narrow age range[54]. Thus, although the current cross-sectional data provide a well-powered insight into differences in white matter microstructure with age, examination of intra-individual change awaits further data. Fully longitudinal studies in larger, wide-age-span samples would be required to track individual trajectories directly, although coverage of a comparable time span to the current study would take many years.

A final limitation pertains to the quality and validity of the water diffusion data. Despite the quality checks on the data conducted both by the UK Biobank imaging team and our own group, we cannot rule out the role of partial volume effects—such as cerebrospinal fluid (CSF) contamination—on the results reported here. Prior reports indicate that MD may be more susceptible to CSF contamination than FA and that this may affect some tracts more than others[12,55]. Thus, it is possible that MD's apparently greater sensitivity to age than FA might be driven by such effects. Whereas tract-averaged data corrected for partial volume effects (using methods such as tract volume[12] or free water elimination[55]) are currently unavailable from UK Biobank, we note that $g$MD and $g$FA were comparably sensitive to age in the context of other brain structural indices (Supplementary Table 15). Although the relatively complex parameterization of the diffusion signal among NODDI measures may be more robust to such effects, the neurobiological validation of these relatively newer parameters (for example, see ref. 56) would benefit from further data. In combination with the variety of cerebrostructural factors that can influence water diffusion discussed above, direct extrapolations of each water diffusion parameter to specific microstructural properties should be made with caution.

This large-scale, single-scanner brain imaging sample has afforded clear insights into the human brain's connections in middle to older age. In this study, we located and quantified the age-related differences in white matter tracts; provided robust information about which diffusion-based biomarkers were especially sensitive in ageing; demonstrated that the inter-individual variation in specific tracts became less specific with age; and found that ageing processes are best modelled as acting on those characteristics that tracts share rather than those that are unique to them. We found evidence to support age-related brain disconnection[1-3] in later life, especially in thalamic and association tracts. In another sense, we found the older brain to be increasingly connected, because its tracts lost some individuality with increasing age. These findings offer secure foundations for planned further exploration of the risk factors and mechanisms of brain and cognitive ageing.

## Methods

**Participants and ethical approval.** The UK Biobank comprises ∼500,000 community-dwelling participants who were initially recruited from across Great Britain between 2006 and 2010, aged 40–69 years (http://www.ukbiobank.ac.uk). An average of 4.15 (s.d. = 0.91) years after initial recruitment, a subset of participants also underwent head MRI at mean age 61.72 (s.d. = 7.47, range 44.64–77.12) years. The initial release of brain dMRI data from 5,455 participants is the subject of the current study. UK Biobank received ethical approval from the research ethics committee (REC reference 11/NW/0382). The present analyses were conducted under UK Biobank application number 10279. All participants provided informed consent to participate. Further information on the consent procedure can be found here (http://biobank.ctsu.ox.ac.uk/crystal/field.cgi?id=200).

**Demographic information.** Information on qualifications, ethnicity, sex and handedness were reported during the initial UK Biobank assessment

(http://biobank.ctsu.ox.ac.uk/crystal/refer.cgi?id=100235). Educational qualifications (UK Biobank code: 6138) were taken from responses to the question: 'Which of the following qualifications do you have? (You can select more than one)'. Response options were as follows: College or University Degree/A levels or AS levels or equivalent/CSEs or equivalent/NVQ or HND or HNC or equivalent/Other professional qualifications, for example, nursing, teaching/None of the above/Prefer not to answer. For the purposes of characterizing the participants here, we collapsed the data into a binary variable, indicating whether or not each participant held a college or university degree. Self-reported ethnic background (UK Biobank code: 21000) was based on response to the question 'What is your ethnic group?'. Response options were as follows: White/Mixed/Asian or Asian British/Black or Black British/Chinese/Other ethnic group/Do not know/Prefer not to answer. These responses were collapsed into White, Mixed and Other. Handedness (UK Biobank code: 1707) was based on responses to the question: 'Are you right or left handed', where response options were as follows: Right-handed/Left-handed/Use both and left equally/Prefer not to answer. At the time of the MRI assessment, participants' medical history (UK Biobank code: 20002) was taken and coded by a trained nurse according to a specific coding tree (http://biobank.ctsu.ox.ac.uk/crystal/field.cgi?id=20002). Those who reported a diagnosis of dementia/Alzheimer's disease or mild cognitive impairment, Parkinson's disease, stroke, other chronic/degenerative neurological problem or demyelinating condition (including multiple sclerosis and Guillain–Barré syndrome) were removed from analysis.

**MRI acquisition.** Details of the image acquisition and processing are freely-available on the UK Biobank website in the form of a Protocol (http://biobank.ctsu.ox.ac.uk/crystal/refer.cgi?id=2367), Brain Imaging Documentation (http://biobank.ctsu.ox.ac.uk/crystal/refer.cgi?id=1977) and in ref. 57. The resultant structural and water diffusion MRI parameters from these processing pipelines were derived by the UK Biobank Imaging team and made available as imaging-derived phenotypes (IDPs). Briefly, all brain MRI data were acquired on a single standard Siemens Skyra 3T scanner with a standard Siemens 32-channel RF receiver head coil, with the imaging matrix angled down by 16° from the AC-PC line. The T1-weighted volumes were acquired in the sagittal plane using a three-dimensional magnetization-prepared rapid gradient-echo sequence at a resolution of $1 \times 1 \times 1$ mm, with a $208 \times 256 \times 256$ field of view. The dMRI protocol employed a spin-echo echo-planar imaging sequence with 10 $T_2$-weighted ($b \approx 0$ s mm$^{-2}$) baseline, 50 $b = 1,000$ s mm$^{-2}$ and 50 $b = 2,000$ s mm$^{-2}$ diffusion-weighted volumes acquired with 100 distinct diffusion-encoding directions and three times multi-slice acquisition. The field of view was $104 \times 104$ mm, imaging matrix $52 \times 52$, 72 slices with slice thickness 2 mm, giving 2 mm isotropic voxels. The flowchart in Supplementary Fig. 1 illustrates the numbers from initial MRI recruitment and attendance through to completion and quality control procedures. Of the 5,455 who provided MRI data, 567 were acquired at an earlier scanning phase (for which the resultant dMRI data are incompatible with data acquired subsequently; see Section 2.10 of the Brain Imaging Documentation). A further 1,314 participants were removed during dMRI quality-control procedures by UK Biobank before data release, which was a combination of manual and automated checking and also included the removal of data badly affected by movement artefacts (as described in the UK Biobank Brain Imaging Documentation). In addition to the 59 participants with self-reported diagnosis of stroke, dementia, Parkinson's disease or any other demyelinating or neurodegenerative disorder, a further two participants with consistently extreme outlying tract-averaged water diffusion biomarker values were removed listwise, along with 35 individual extreme outlying data points (<0.001% of total data), following visual inspection of the data by the authors, leaving a total of 3,513 participants for analysis in the current study. The current sample did not differ from those not scanned with respect to age at initial recruitment ($t = 0.834$, $P = 0.405$), but comprise a higher proportion of females ($\chi^2 = 9.637$, $P = 0.002$). The total numbers of available tracts are reported in Supplementary Tables 1 and 2.

**Diffusion MRI processing and tractography.** FA and MD are commonly derived variables, which describe the directional coherence and magnitude of water molecule diffusion, respectively. Water molecules tend to diffuse with greater directional coherence and lower magnitude when constrained by tightly packed fibres (such as well-myelinated axons) and by cell membranes, microtubules and other structures[26]. Thus, individual differences in FA and MD in brain white matter reflect meaningful differences in underlying microstructure, borne out by comparison with brain white matter post-mortem work[4,5]. Measures of tract-averaged $\lambda$ax and $\lambda$rad and MO were also available as IDPs from UK Biobank. The former two measures are also parameters of interest to brain ageing (for example, refs 8,10) but are similar in their derivation from the three main tensor eigenvalues: MD is the mean of all three, whereas $\lambda$ax = $\lambda$1 and $\lambda$rad is the mean of $\lambda$2 and $\lambda$3). MO (also known as the mode of anisotropy) describes the third moment of the tensor (a positive value denotes narrow tubular water diffusion, whereas a negative number denotes planar water diffusion), although little work has been done to examine MO in relation to ageing. Consequently, we also provide parallel analyses of $\lambda$ax, $\lambda$rad and MO in Supplementary Material.

Unlike standard diffusion tensor MRI, NODDI makes specific assumptions about the way in which local microarchitecture affects the geometric diffusion

of water and parameterizes the water diffusion signal according to one of three geometrical models: free water diffusion (such as in CSF), restricted diffusion caused by the presence of dendrite and axons bodies, and hindered diffusion among cell bodies. The resultant indices describe the ICVF (a measure of neurite density), ISOVF (a measure of extracellular water diffusion) and neurite OD (the degree of fanning or angular variation in neurite orientation).

Gradient distortion correction was applied using tools developed by the Freesurfer and Human Connectome Project groups, available at https://github.com/Washington-University/Pipelines. The Eddy tool from FSL (http://fsl.fmrib.ox.ac.uk/fsl/fslwiki/EDDY) was then used to correct the data for head motion and eddy currents. Next, within-voxel multi-fibre tract orientation structure was modelled using BEDPOSTx followed by probabilistic tractography (with crossing fibre modelling) using PROBTRACKx[58–60]. Automatic mapping of the 27 major white matter tracts was conducted in standard space of each participant using start/stop region of interest masks (implemented using the AutoPtx plugin for FSL)[61] to derive tract-averaged measures of FA and MD for the following tracts of interest: MCP, FMaj, FMin and bilateral medial lemnisci, CSTs, acoustic, anterior thalamic, posterior thalamic, STRs, superior, inferior longitudinal and inferior fronto-occipital fasciculi, and both the cingulate gyrus and parahippocampal portions of the cingulum bundle (Fig. 1 in the main document). In addition, NODDI[24] modelling of the dMRI data was conducted using the AMICO tool (Accelerated Microstructure Imaging via Convex Optimization; https://github.com/daducci/AMICO)[62]. Maps of ICVF, ISOVF and OD, registered with the AutoPtx tract masks, allowed the calculation of tract-averaged values for each parameter across all voxels pertaining to each tract of interest.

**Volumetric MRI processing.** Extraction of the brain was achieved by nonlinearly warping the data to MNI152 space (FNIRT)[63,64], with the brain mask then back-transformed into native space. FAST[65] was then used to segment the brain tissue (in native space, to avoid noise due to interpolation) into the CSF, grey matter and white matter (with total brain volume being the sum of grey and white matter volume); bilateral hippocampal and thalamic volumes were derived using FIRST[66]. All volumes were then adjusted for head size using a SIENAX-style analysis[67]. This involves deriving a scaling factor from the normalization transform matrix obtained from the affine registration of skull tissue between T1-weighted volume and MNI152 space. The resultant scaling factor was then applied to the volumes of interest for each participant.

**Statistical analysis.** Statistical analyses were conducted using R v3.2.2 (Fire Safety) and MPlus v7.3 (ref. 68). The distribution of each white matter tract diffusion measure was inspected and we found no large deviations from normality. We briefly illustrate the advantage in precision that this large sample affords, by conducting a comparison of the minimal detectable difference that can be achieved between the current sample size and a smaller study. A study with 1,000 participants would have 80% power to detect a bivariate association at $\alpha = 0.05$ if the true effect is $\beta_{standardized} = 0.088$. In contrast, the present study ($n = 3,513$) has the same power to detect a $\beta_{standardized} = 0.047$ (calculated using G*Power version 3.0.10; http://www.gpower.hhu.de/en.html).

Handedness is sometimes reported in dMRI studies, under the assumption that differences exist (for example, Bender et al.[8]). There were no substantive differences in tract characteristics between left- and right-handed participants (except for a trend for left-handers to have marginally higher FA ($t(378.53) = 2.130$, $P = 0.034$, $d = 0.219$) and lower OD ($t(374.58) = -2.747$, $P = 0.006$, $d = 0.141$) in the FMin, lower OD in the left CST ($t(378.19) = -2.446$, $P = 0.015$, $d = 0.125$) and higher OD in the right Uncinate ($t(373.85) = 2.297$, $P = 0.022$, $d = 0.238$). As a result, all analyses are reported across the entire group. Initially, associations between FA and MD of specific tracts were modelled with respect to age, sex and hemisphere using multiple regression. We also included an age × sex interaction term to examine whether age-related trajectories differed for men and women. Linear and quadratic models were compared and, where models with an $age^2$ term exhibited a significantly better fit ($P < 0.05$), these results were reported. Owing to the large number of comparisons, a threshold of $P < 0.001$ was used to denote significant effects within each model. The age and $age^2$ components of these models were illustrated for each tract and water diffusion parameter using kernel density plots, to allow clearer visualization of data point overlap than is possible with standard scatter plots with a sample of this size. Tests of the difference between association magnitudes were conducted on the linear component of tract measures, implemented using Williams's $t$ for dependent groups with overlapping correlations (cocor.indep.groups.overlaps) as implemented in the cocor R package (http://cran.r-project.org/web/packages/cocor/cocor.pdf).

Confirmatory factor analysis was used to produce one-factor models for each of the five white-matter microstructural measurements (FA, MD, ICVF, ISOVF and OD). On the basis of the first model, for the measurement of FA, we excluded five tracts that had low ($<0.3$) loadings on the general factor: the MCP and bilateral ML, and parahippocampal cingulum. For consistency, we also removed these tracts from the subsequent factor models. To improve the model fit for the subsequent use of the factors and the subsequent models, we used the modification indices option in the Mplus v7.3 package[68] to add additional, residual covariances (paths linking specific tracts to one another). Importantly, none of the results of the study was substantially altered by dropping these residual covariances, or by

including the four tracts that were removed. All the models adjusted each tract for sex, age and—for all measurements other than FA, for which the tracts did not show many appreciable quadratic age curves—$age^2$. Fit statistics for the factor models are presented in Supplementary Table 5, standardized factor loadings are provided in Supplementary Table 6 and a list of the residual covariances can be found in Supplementary Table 7. Factor score estimates were extracted from these models and used in further analyses, and the one-factor model provided the basis for the age-moderation models described below.

We also used structural equation modelling to test whether the age variation in the tracts was best represented as a common pathway model (where age has associations with only the latent factor), an independent pathways model (where age has associations with each of the individual tracts separately and not with the latent factor) or a common + independent pathways model (where age is associated with the latent factor and also with some specific factors, as required)[69]. To estimate the common + independent pathway model, we first included a path from age (which had been centred before inclusion in the model) to the latent factor, then added those paths that produced statistically significant improvements in model fit. For MD, ICVF, ISOVF and OD, we also included paths from $age^2$ alongside every significant age path. We always considered the paths together; if the age path was added, so was the $age^2$ path. We then used model fit indices ($\chi^2$-test, Akaike Information Criterion and Bayesian Information Criterion) to examine the fit differences between the three models. Fit statistics are shown in Supplementary Table 9. In all cases, the common pathways model was the worst fitting. For MD, ICVF, ISOVF and OD, the common + independent pathway model fit either significantly better or no different to the independent pathway model; for FA, there was some ambiguity, with Bayesian Information Criterion (BIC) demonstrating better fit for the common + independent pathways model but Akaike Information Criterion (AIC) demonstrating poorer fit.

We examined the degree to which the effect of increasing age on generally lower water diffusion coherence (FA) across tracts was attributable to lower neurite density (ICVF) and/or the amount of tract complexity/fanning (OD) using a multiple mediator model in a structural equation model framework[70] in the lavaan R package[71]. Age was set as the X (independent) variable, the factor score for FA (gFA) was set as the Y (outcome) variable, with gICVF and gOD as two covarying mediators (M). The degree to which the association between X and Y (known as the c path) is attenuated by M (the c' path) denotes the mediation effect[72,73].

Tests of the relative explanatory power of the factor scores of all five water diffusion parameters to account for age variance—beyond measures more conventionally associated with ageing (total brain, grey matter, white matter and bilateral hippocampal volumes)—was achieved using penalised (elastic net) regression[74] bootstrapped 1,000 times. This method allowed us to identify an optimal combination of predictors from among a group of highly collinear variables. We randomly split the participants into two equal halves (Train and Test). The optimal predictor set was identified by running penalized elastic net regression in the Train group, selecting only those measures that were identified in the majority ($>60\%$) of models. A multiple linear regression was then conducted in the Train set and a confirmatory multiple linear regression in the Test set. In general, if the model $\beta$ and $R^2$ are comparable, the combination of identified predictors are considered optimal for the whole sample.

A posteriori, we also examined the relative contribution of the volume of the thalamus and the microstructure of its radiations (gTR; the first unrotated solution of a principal components analysis of the anterior, superior and PTRs) to age variance, beyond global atrophy (total brain volume, corrected for head size). We did so by running two sets of multiple regressions with age as the outcome variable. In the first, the volume of the thalamus and gTR for each of FA, MD, ICVF, ISOVF and OD were covariates. In the second, we added total brain volume corrected for head size, to ascertain whether these main effects were simply reflective of global atrophy.

White matter tract de-differentiation was initially explored by visualizing whether the strength of cross-tract associations altered across age groups for FA, MD, ICVF, ISOVF and OD separately for all tracts. Six age groups were created with ~5 year intervals. These were: 44.64–49.98 years ($n = 290$, $M = 48.23$, s.d. = 1.16); 50.05–54.99 years ($n = 505$, $M = 52.61$, s.d. = 1.43); 55.00–60.00 years ($n = 559$, $M = 57.57$, s.d. = 1.50); 60.01–65.00 years ($n = 800$, $M = 62.57$, s.d. = 1.45); 65.00–70.00 years ($n = 876$, $M = 67.38$, s.d. = 1.38); 70.00–77.12 ($n = 483$, $M = 72.45$, s.d. = 1.72). To more formally quantify this effect, age moderation models were estimated separately for each of the white matter microstructural measurements (that is, five models, one for each of FA, MD, ICVF, ISOVF and OD). The age-moderation models included all of the residual covariances that were added to the one-factor models as described above. In the first set of models, we extended each of the five general factor models to include an interaction parameter representing age moderation of the shared variance across the 22 tracts and an interaction parameter representing age moderation of the tract-specific unique variance components. In other words, we estimated two interaction parameters for each of the five diffusion measures. This model can be written for diffusion measure Y in tract $t$ as:

$$Y[t]_n = \nu[t] + \alpha_1[t] \times age_n + (\alpha_2[t] \times age_n^2) + \alpha_3[t] \times sex_n \\ + (\lambda_1[t] + \lambda_1'[t] \times \lambda_1[t] \times age_n) \times gY_n + (\lambda_2[t] + \lambda_2'[t] \times \lambda_2[t] \times age_n) \times uY[t]_n \quad (1)$$

where $\nu[t]$ is a tract-specific regression intercept; $\alpha_1[t]$, $\alpha_2[t]$ and $\alpha_3[t]$ are tract-specific regression coefficients for the effects of age, $age^2$ (not included for FA) and sex; $\lambda_1[t]$ is a tract-specific loading (main effect) on the general factor (gY);

$\lambda_2[t]$ is a tract-specific loading (main effect) on the tract-specific unique factor ($uY[t]$); and $\lambda_1'$ and $\lambda_2'$ are tract-invariant interaction parameters representing moderation of the loadings on the general factor and the tract-specific unique factors, respectively. The subscript $n$ indicates that a variable varies across individuals. In the above equation, the interaction terms are multiplied by the tract-specific loading in addition to the corresponding latent factor, to specify age moderation to occur proportionally to the magnitude of the tract-specific loadings (see Cheung et al.[75]).

We calculated communality values (proportion of variance explained by the general factor relative to total variance (variance explained by the general factor plus residual variance)[76], Appendix B) and their s.e. for 5-year increments of the sample's age range (that is, for ages 45, 50, 55, 60, 65, 70 and 75 years). Next, using cubic polynomial interpolation, we calculated the expected values at all other ages between 45 and 75 along with their s.e. and converted these into the age trajectories with 95% confidence intervals shown in Fig. 8 in the main document.

To evaluate whether the overall pattern of increasing communality with age was driven by smaller subset of the tracts, we estimated more complex models that estimated tract-specific interaction parameters representing age moderation of loadings on the general factor and tract-specific interaction parameters representing age moderation of tract-specific uniquenesses[77]. In other words, we estimated 44 interaction parameters (1 interaction for the loading on the general factor and 1 interaction for the unique component, for each of the 22 tracts) for each of the 5 diffusion measures (FA, MD, ICVF, ISOVF and OD). This model can be written as

$$Y[t]_n = \nu[t] + \alpha_1[t] \times age_n + \left(\alpha_2[t] \times age_n^2\right) + \alpha_3[t] \times sex_n \\ + \left(\lambda_1[t] + \lambda_1[t]' \times age_n\right) \times gY_n + \left(\lambda_2[t] + \lambda_2[t]' \times age_n\right) \times uY[t]_n \quad (2)$$

where the interaction terms $\lambda_1'$ and $\lambda_2'$ are estimated individually for each tract, as indicated by the suffix $[t]$. As interaction terms are estimated individually for each tract, the proportionality constraint (achieved in equation (1) via multiplication by the tract-specific loading) is not necessary.

Within the de-differentiation models, we tested whether the age moderation for each loading and each uniqueness was statistically significantly different from zero (that is, whether the parameters $\lambda_1'$ and $\lambda_2'$ differed significantly with age). The results of these tests are provided in Supplementary Tables 14 and 15 for loadings and uniquenesses, respectively. To calculate the communality for each tract (that is, the proportion of the total variance in that tract explained by the factor), we divided the age-specific shared variance in that tract by the age-specific shared-plus-unique (that is, total) variance in that tract[76] (Appendix B), resulting in the age trend for the communalities shown in the right panels of Supplementary Figs 10–16.

Finally, local structural equation modelling[78,79] was used to provide a non-parametric confirmation of the multi-parameter models described above. Using batch running of Mplus models[80], we ran 300 one-factor models for each diffusion measure, each aimed at a different part of the age range (300 equal increments between 45 and 75 years). From the outputs of these models, we plotted the factor loadings and uniquenesses as a function of age, for each tract. Again, we also calculated the communality for each tract. The graphical outputs from the local structural equation modeling models are shown in Supplementary Figs 10–16: they confirm the results from the multi-parameter models, providing a more detailed view of the precise age trajectories.

**Data availability.** All data analysed herein (including IDPs) were provided by UK Biobank under project reference 10279, subject to a data transfer agreement. Researchers can apply to use the UK Biobank data resource for health related research in the public interest. A guide to access is available from the UK Biobank website (http://www.ukbiobank.ac.uk/register-apply/).

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

## Acknowledgements

We thank the UK Biobank participants for their participation and the UK Biobank team for their work in collecting, processing and disseminating these data for analysis. This research was conducted, using the UK Biobank Resource under approved project 10279, in The University of Edinburgh Centre for Cognitive Ageing and Cognitive Epidemiology (CCACE) (http://www.ccace.ed.ac.uk), part of the cross-council Lifelong Health and Wellbeing Initiative (MR/K026992/1). Funding from the Biotechnology and Biological Sciences Research Council (BBSRC) and Medical Research Council (MRC) is gratefully acknowledged. S.R.C. was supported by MRC grant MR/M013111/1. I.J.D; S.J.R., D.C.L., S.P.H. and G.D. are supported by the Medical Research Council award to CCACE (MR/K026992/1). I.J.D. is additionally supported by the Dementias Platform UK (MR/L015382/1), and he and S.R.C. by the Age UK-funded Disconnected Mind project (http://www.disconnectedmind.ed.ac.uk). S.J.R. was partly supported by a grant from Boehringer Ingelheim Fonds. E.M.T.-D. was supported by National Institutes of Health (NIH) research grants HD083613, HD081437 and AA023322. E.M.T.-D. is a member of the Population Research Center at the University of Texas at Austin, which is supported by NIH centre grant HD042849. E.M.T.-D. was also supported as a Visiting Scholar at the Russell Sage Foundation. J.M.W. was supported by the Scottish Imaging Network: A Platform for Scientific Excellence (SINAPSE) collaboration (http://www.sinapse.ac.uk).

## Author contributions

S.R.C. and S.J.R. drafted the manuscript and conducted statistical analyses. E.M.T.-D., M.E.B. and I.J.D. supported statistical analysis and edited the manuscript. D.C.L. and S.P.H. provided data support. S.P.H., G.D., J.M.W. and C.R.G edited the manuscript.

## Additional information

**Competing financial interests:** The authors declare no competing financial interests.

