## [Peer Review File · Nature Communications]

Editorial Note: this manuscript has been previously reviewed at another journal that is not operating a transparent peer review scheme. This document only contains reviewer comments and rebuttal letters for versions considered at Nature Communications. Mentions of prior referee reports have been redacted.

Reviewers' comments:

Reviewer #1 (Remarks to the Author):

The authors describe associations between age and white matter microstructure in a sample of 3,513 adults aged between 45 and 77 years old. Mean diffusivity was found to be the most age-sensitive diffusion measure, and inter-individual differences in white matter microstructure across brain tracts were shown to be increasingly correlated in older age. The sample size and de-differentiation hypotheses add originality and interest to the study, the paper is well written and presented and comes with comprehensive supplementary material. The authors have responded to all my previous suggestions, and I have no further comments.

Reviewer #2 (Remarks to the Author):

I have reviewed this article [redacted] and I really appreciate the fact that my comments as well as the authors' responses were included in this submission to Nature Communications. This greatly facilitates the job of a reviewer.

I am mostly satisfied with all answers the authors provided, and I would like to congratulate them on doing such a thorough job.

However, I still think that the authors could include a more qualitative illustration of the de-differentiation effect in the main text. Their main figures at the moment are: Fig1: 3D rendering of tracts - to me belongs to supplementary info, but I agree they are pretty; Figs2-4 all show more or less the same thing but displayed in different ways; importantly this is the less novel part of the results. Fig5 is the only figure that shows the novel de-differentiation results. I agree that there are already very many detailed figures in the supplementary material - I just find it odd that the main text's figures put so much emphasis on the less novel results and so little emphasis on the more novel results.

Reviewer #3 (Remarks to the Author):

I concur with the comments from Reviewer #2 that the novel finding presented here is in Figure 5, showing de-differentiation of white matter pathways in ageing. The previous 4 figures mainly show effects that have already been reported in previous studies. I also

concur with Reviewer 2's suggestion that it would be helpful to have a more intuitive figure demonstrating this finding. As reviewer 2 notes, it is not a question about whether the results are statistically significant but, rather, allowing the reader to appreciate whether it is a small or large effect. Visualizing correlations for age bins might be a way to do this as changes in correlation coefficients are readily interpretable and will also lend themselves to power calculations that will be useful for future studies. I understand the authors' hesitance to pursue this simple analysis but the results might be informative and easy to interpret.

Other comments:

-They don't seem to present any clear evidence either in favor of or against the last-in-first-out hypothesis. To test this hypothesis they would need to measure development so that they could ascertain the age at which the tracts develop. These claims should be removed unless there is data to support the claims.

-The Methods should clarify whether tractography was conducted in each individual brain or only on an average brain. If the calculations are done based on the average location of the tracts (rather than the tracts calculated in the individual brains) then there is a major confound of tract size. For a subject with a smaller tract, there will be much more partial voluming than in a subject with a larger tract.

Responses to Reviewer Comments

Reviewer #1

The sample size and de-differentiation hypotheses add originality and interest to the study, the paper is well written and presented and comes with comprehensive supplementary material. The authors have responded to all my previous suggestions, and I have no further comments.

We thank the reviewer again for their helpful comments and are glad that we have entirely addressed their concerns.

Reviewer #2

I am mostly satisfied with all answers the authors provided, and I would like to congratulate them on doing such a thorough job.

We thank the reviewer for their kind words and additional useful comments.

However, I still think that the authors could include a more qualitative illustration of the de-differentiation effect in the main text. Their main figures at the moment are: Fig1: 3D rendering of tracts - to me belongs to supplementary info, but I agree they are pretty; Figs2-4 all show more or less the same thing but displayed in different ways; importantly this is the less novel part of the results. Fig5 is the only figure that shows the novel de-differentiation results. I agree that there are already very many detailed figures in the supplementary material - I just find it odd that the main text's figures put so much emphasis on the less novel results and so little emphasis on the more novel results.

We respect that Reviewers #2 and #3 feel strongly about this, and have now added an additional main figure as a precursor to the detailed de-differentiation analysis, in order to illustrate the phenomenon of interest. These are now presented as main Figure 5. We have also altered the text as follows:

Methods:

“White matter tract de-differentiation was initially explored by visualising whether the strength of cross-tract associations altered across age groups for FA, MD, ICVF, ISOVF and OD separately. Six age groups were created with approximately 5 year intervals. These were: 44.64 - 49.98 years (n = 290, M = 48.23, SD = 1.16); 50.05 - 54.99 years (n = 505, M = 52.61, SD = 1.43); 55.00 - 60.00 years (n = 559, M = 57.57, SD = 1.50); 60.01 - 65.00 years (n = 800, M = 62.57, SD = 1.45); 65.00 - 70.00 years (n = 876, M = 67.38, SD = 1.38); 70.00 - 77.12 (n = 483, M = 72.45, SD = 1.72). In order to more formally quantify this effect, age moderation models were estimated...”

Results (p. 11)

“A series of heatmaps (Figure 5) illustrate the increasing relatedness of white matter tract microstructure across 6 age groups with approximately 5 year intervals. Qualitatively stronger associations among tracts with increasing age were evident for FA, MD, ICVF and ISOVF, but not for OD. To quantify this more formally, we extended...”

Reviewer #3

I concur with the comments from Reviewer #2 that the novel finding presented here is in Figure 5, showing de-differentiation of white matter pathways in ageing. The previous 4 figures mainly show effects that have already been reported in previous studies. I also concur with Reviewer 2's suggestion that it would be helpful to have a more intuitive figure demonstrating this finding. As reviewer 2 notes, it is not a question about whether the results are statistically significant but, rather, allowing the reader to appreciate whether it is a small or large effect. Visualizing correlations for age bins might be a way to do this as changes in correlation coefficients are readily interpretable and will also lend themselves to power calculations that will be useful for future studies. I understand the authors' hesitance to pursue this simple analysis but the results might be informative and easy to interpret.

We are grateful for the reviewer's additional comments. This point is well-taken (see also comment and response from Reviewer #2), and we have now added these illustrations as Figure 5.

Other comments:

-They don't seem to present any clear evidence either in favor of or against the last-in-first-out hypothesis. To test this hypothesis they would need to measure development so that they could ascertain the age at which the tracts develop. These claims should be removed unless there is data to support the claims.

We thank the reviewer for pointing this out – we agree it is important to explicitly test whether the age associations are larger in some classes of tract than in others. We now provide this analysis in the results section:

“Differences in the magnitude of associations with age between these thalamic and association fibres (IFOF, ILF, SLF, Uncinate, ATR, STR and PTR) were consistently and significantly greater than those for the cingulum bundles, CST, MCP, ML, FMaj, AR (Williams's t-values > 2.42, p-values < 0.015) across FA, MD, ICVF and ISOVF, with the exception of the CingG for ICVF.”

We also agree that the present data cannot provide insights into white matter development. However, this should not preclude us from relating our work to other data from earlier life in the form of post mortem and dMRI investigations (or publications wherein these data are discussed). We cite each of these type of reference to substantiate our point (Discussion, 3rd and 4th paragraphs). Though this section is comparatively brief, we judge that it is important that we interpret this pattern of regional sensitivity to age.

-The Methods should clarify whether tractography was conducted in each individual brain or only on an average brain. If the calculations are done based on the average location of the tracts (rather than the tracts calculated in the individual brains) then there is a major confound of tract size. For a subject with a smaller tract, there will be much more partial voluming than in a subject with a larger tract.

We have now made it clearer that the tractography itself was conducted in native space for each individual (Methods, page 22).

REVIEWERS' COMMENTS:

Reviewer #1 (Remarks to the Author):

The authors have addressed my comments